# A Comparative Study of Skeletal and Dental Outcome between Transcutaneous External Maxillary Distraction Osteogenesis and Conventional Rigid External Device in Treating Cleft Lip and Palate Patients

**DOI:** 10.3390/jpm12071062

**Published:** 2022-06-29

**Authors:** Chi-Yu Tsai, Yi-Hao Lee, Te-Ju Wu, Shiu-Shiung Lin, Jui-Pin Lai, Yu-Jen Chang

**Affiliations:** 1Department of Craniofacial Orthodontics, Kaohsiung Chang Gung Memorial Hospital, College of Medicine, Chang Gung University, No. 123, Dapi Rd., Niaosong Dist., Kaohsiung City 83340, Taiwan; charlie-dark@yahoo.com.tw (C.-Y.T.); lkkk05@hotmail.com (Y.-H.L.); orthowilliam@gmail.com (T.-J.W.); glasgow1993@yahoo.com (S.-S.L.); 2Department of Dentistry, Kaohsiung Chang Gung Memorial Hospital, College of Medicine, Chang Gung University, No. 123, Dapi Rd., Niaosong Dist., Kaohsiung City 83340, Taiwan; 3Department of Plastic and Reconstructive Surgery, Kaohsiung Chang Gung Memorial Hospital, College of Medicine, Chang Gung University, No. 123, Dapi Rd., Niaosong Dist., Kaohsiung City 83340, Taiwan; benjplai@yahoo.com

**Keywords:** cleft lip and palate, distraction osteogenesis, rigid external device, transcutaneous maxillary distraction, long-term stability, soft tissue profile

## Abstract

Background: Traditional distraction osteogenesis (DO) with the tooth-borne rigid external device (RED) system was regularly used in treating patients with cleft-related maxillary hypoplasia. However, the bone-borne RED system with miniplates and bone screws has currently become an effective treatment. This retrospective study was to compare bone-borne RED with traditional tooth-borne RED in distraction effectiveness, blood loss, operative time, and long-term stability. Methods: Twenty-two growing patients who underwent RED therapy were divided into two groups: eleven patients utilizing the bone-borne RED system with the transcutaneous wire attached with skeletal anchorage; another eleven patients using the traditional tooth-borne RED system with the intra-oral device attached with dental anchorage. Serial lateral cephalograms were analyzed for comparing treatment outcomes and stability in 1 month, 6 months, and 1.5 years after distraction. Results: In bone-borne RED group, the maxilla was advanced by 19.98 mm with slight clockwise rotation of 0.40° and minimal palatal inclination change of incisor by −3.94°. In traditional tooth-borne RED group, the maxilla showed less advancement by 14.52 mm, with significant counter-clockwise rotation of −11.23° and excessive palatal inclination change of incisor by −10.86°. Although operative time was longer in the bone-borne RED group by 38.4 min, this did not bring about greater blood loss. Conclusions: the bone-borne RED via transcutaneous wire system provides an easy, simple, and comfortable procedure as well as favorable long-term stability in maxillary distraction.

## 1. Introduction

Cleft-related maxillary hypoplasia is a congenital maxillofacial deformity characterized by facial growth disturbance, poor facial esthetic, improper occlusion, and negative psychological impact. Maxillary advancement osteotomy is a widely-used surgical option in treating patients with midface hypoplasia. Conventional LeFort I osteotomy involves significant maxillary forward repositioning to achieve the desired pre-planned position to restore normal jaw function and facial esthetic in patients with cleft lip and palate (CL/P) [1]. Unfortunately, the conventional orthognathic surgery can only be carried out at the completion of growth to ensure a long-term stable result, the young CL/P patients should wait for skeletal maturity to adopt the surgical procedure. Besides, the soft tissue tension derived from lip scar, palatal flap contracture, and excessive skin stretch may contribute to the backward relapse of maxilla after LeFort I advancement [2].

Maxillary distraction osteogenesis (DO) has been developed as an innovative surgical option for correcting cleft-related maxillary retrusion at an early stage of life. The DO in the craniofacial region was first introduced by McCarthy for lengthening the retrognathic mandible by 18–24 mm in young patients with hemifacial microsomia and Nager’s syndrome [3]. In 1997, Polley and Figueroa developed a tooth-borne rigid external distraction device (RED) in combination with intra-oral splint for advancing the retrusive maxilla in CL/P patients. Since then, the RED procedure has become a popular alternative approach for maxillary DO, with minimal skeletal relapse and excellent esthetic outcome [4].

The original design of the tooth-borne RED contains an external distractor and prefabricated intra-oral splint [4]. The intra-oral splint is modified from headgear orthodontic appliances with vertical projecting arms extending from the oral cavity, and the height of the arms should be meticulously determined for better distraction direction. There are some drawbacks when using the tooth-borne RED approach. First, the fabrication of a custom-made intra-oral appliance requires considerable laboratory time and expertise. Second, patients should provide adequate and healthy dentition for applying an intra-oral splint to attach to the distractor. Third, the use of a tooth as an anchor for maxillary distraction would result in significant dentoalveolar changes rather than bony movements [5]. Hence, those disadvantages indeed restrict the clinical indication for choosing tooth-borne RED technique as an alternative treatment.

With the invention of innovative orthopedic material, bone plates, or screws are recommended to the replace intra-oral splint to directly connect the external distractor and maxillary bone. Hierl and Hemprich carried out a midface DO by using miniplates as skeletal anchorage in treating an edentulous adult CL/P patient [6]. Monaghan et al. successfully treated 10 cases that underwent the external distraction method by using miniplates with transcutaneous wire [7]. Jenny presented an alar pinning technique in rigid external distraction for treating 7 patients, and all of them had stable and significant maxillary projection with the aid of bone-borne devices [8]. Zheng proposed an internasal bone-borne traction hook to replace the fixation plate for external distraction, and showed good final outcomes [9]. The bone-borne RED turned out to be an effective, easy, and comfortable alternative method.

Numerous studies have suggested that the introduction of the tooth-borne RED approach would allow surgeons to treat patients with cleft-related maxillary hypoplasia [2,4,5]. However, no studies have reported the maxillary distraction efficacy of the innovative bone-borne RED system with skeletal anchorage in comparison with the traditional tooth-borne RED approach with dental anchorage to date. The current study demonstrated the novel DO procedure by using bone-borne RED with transcutaneous wire system, and then compared the distraction efficacy, volume of blood loss, operative time, and long-term distraction outcomes between the bone-borne RED system and the traditional tooth-borne RED approach.

## 2. Materials and Methods

### 2.1. Subjects

This study enrolled 22 growing patients with cleft-related maxillary hypoplasia who underwent maxillary DO procedure between April 2012 to July 2020 at the craniofacial center of Kaohsiung Chang Gung Memorial Hospital. The patient’s data were collected from their medical charts, and the variables included demographic characteristics (gender, age), duration of operation (minutes) and distraction (days), perioperative estimated blood loss (mL), and measurements of cephalometric analysis. The exclusion criteria were those with syndromic CL/P or other types of congenital craniofacial deformities requiring extensive reconstructive consideration; and simultaneous adjunctive surgical management such as multi-segmental osteotomies and adjunctive cosmetic soft tissue surgery.

A total of 22 CL/P patients included in this study (15 male and 7 female; mean age: 11.6 ± 0.6 years; range: 10.7–13.2) were divided into two groups based on the distraction method employed. The bone-borne RED group consisted of 11 patients (7 men and 4 women; mean age: 11.5 ± 0.7 years; range: 10.7–13.2) whose retruded maxilla was distracted forward through transcutaneous wires that attached directly onto the maxillary bone surfaces. The traditional tooth-borne RED group composed of 11 patients (8 male and 3 female; mean age: 11.8 ± 0.5 years; range: 11.1–12.9) who underwent maxillary distraction by using prefabricated intra-oral splint that connected to the dentition and external distractor.

All the patients underwent presurgical computed tomography (CT) scanning using a 64-slice Toshiba Aquilion 64 system (Toshiba Medical Systems, Otawara, Japan) in the supine position, and the standardized lateral cephalograms were also made in the reproducible standing natural head position. The radiographic images provided important information for evaluating inherent craniofacial deformities, detecting the bony defects, determining the final desired maxillary position, and assessing the distraction outcomes.

### 2.2. Presurgical Dental Preparation for the Tooth-Borne RED Group

Pre-surgical dental preparation was only carried out with the traditional tooth-borne RED method for fabricating intra-oral appliance. The dental impression was taken at chairside, and the intra-oral splint was made by a senior orthodontist through a laboratory procedure. The intra-oral splint was fabricated from the modification of headgear facebow. The inner bows of the facebow were fixed onto the maxillary molar bands. The outer bows were bent over to form two extra-oral vertical projecting arms, and the ends of the arms were then bent into circle-shaped traction eyelet at the level of alar base. The patients were called back to try on the custom-made intra-oral splint at chairside; thereafter, the intra-oral splint was equipped one day before surgery (Figure 1A,B).

### 2.3. Surgical Technique

Under general anesthesia, all patients underwent buccal sulcus incision, LeFort I osteotomy with pterygomaxilliary dysjunction and down-fracture procedure, and the RED II device (KLS-Martin L.P, Tuttlingen, Germany) was installed by the same craniofacial surgeon (J.P Lai). The RED head frame was secured to the cranium by penetrating three pins bilaterally into scalp, and was oriented parallel to the Frankfurter horizontal plane.

Following exposure of the maxillary bone surface, the bone-borne RED group incorporated two anchoring miniplates that were inserted bilaterally onto the medial buttresses of the mobilized maxilla. A total of five bone screws were used to stabilize the two miniplates, and one screw hole of each miniplate was left for applying the transcutaneous stainless-steel wires, and subsequently, the wire passing out through lateral alar base via skin incision. The intra-oral incisions were closed afterwards (Figure 2A,B).

### 2.4. Distraction Methods

Maxillary DO was started after a latency period of 3−15 days (bone-borne RED group: 8.55 ± 3.05 days; traditional tooth-borne RED group: 8.09 ± 0.83 days) after surgery. In the bone-borne RED group, the transcutaneous wires were simply attached to the external distractor, and the distraction force and vector were carried out directly onto the maxilla via tightening the transcutaneous wires, which were also attached to anchoring miniplates on maxillary bone surfaces (Figure 2D,G). In the traditional tooth-borne RED group, maxillary distraction was initiated by tightening the stainless-steel wires that were connecting between the vertical projecting arms of the intra-oral splint and the extra-oral distractor (Figure 1D,G). The height of the projecting arms is of crucial importance to control the force vector and rotational movement during maxillary distraction. Distraction was commenced at a rate of 1 mm per day with a rhythm of 1 turn/0.5 mm twice a day, normally one turn in the morning and once at night. The duration of distraction was calculated until the desired maxillary advancement was achieved (bone-borne RED group: 31.1 ± 9.9 days; traditional tooth-borne RED group: 35.6 ± 11.5 days), and the traction vector modification was meticulously evaluated and adjusted once a week. The RED device was retained without activation for consolidation phase (bone-borne RED group: 69.0 ± 11.5 days; traditional tooth-borne RED group: 72.9 ± 20.9 days), and then the RED device and anchoring miniplates were removed under general anesthesia with outpatient clinical procedure.

### 2.5. Blood Loss and Operation Time Assessment

The estimated blood loss (EBL) for each surgery was assessed by weighing the sponges and measuring the suction volume, in accordance with standard operation theater procedures. The duration of the operation was measured from the start of the incision to the completion of the last suture [10].

### 2.6. Outcome Assessment

The distraction outcomes were assessed by comparing the movements of selected skeletal, dental, and soft tissue landmarks on cephalograms taken 1 week before (T0), termination of DO (T1), 6 months after (T2), and 1.5 years after (T3) distraction. The 11 anatomical landmarks and 24 parameters were chosen, and their definitions were shown in the Appendix A Appendix A. The definitions of all the landmarks and reference lines were derived and modified from Huston et al. [11] and Lin et al. [12].

The AudaxCeph Empower software (VER5.2, Ljubljana, Slovenia) was used to analyze the changes of parameters among the above 4 stages (T0-T3). An x-y coordinate system was designed to measure the displacement of selected landmarks in two dimensions on the serial cephalograms. A horizontal reference line approaching the true horizontal line was constructed 7 degrees below the sella–nasion (S-N) line as the *X*-axis [13]. Then the *Y*-axis was determined by drawing a vertical reference line perpendicular to the *X*-axis by intersecting the line at the Sella point. Therefore, the assessment of linear and angular parameters derived from each landmark on cephalograms could be measured according to the X-Y coordinate system, and the measurements were recorded between each phase by a single examiner (C.Y Tsai) (Figure 3A,B). Table 1 illustrated the definition of linear parameters in relation to the *X*-, *Y*-axis.

### 2.7. Statistical Analysis

The data were statistically analyzed using the SPSS software program, version 20.0 (SPSS Inc., Chicago, IL, USA). The Kolmogorov–Smirnov test was used to test the normality of data distribution. The Paired Sample *t*-test was used for analyzing the changes of parameters between T0-T1, T1-T2 and T2-T3 within each group; and Independent Sample *t*-test was used to compare the means of two independent groups. All variables were presented as mean ± SD, and the level of statistical significance was set at *p*-value < 0.05.

A single examiner repeated six angular (SNA, SNB, SN-MP, SN-PP, U1-SN, U1-PP) and six linear (ANS-x,y, A-x,y, U1i-x,y) measurements from eleven of twenty-two randomly selected patients in 2 occasions with 1-month interval to confirm the reproducibility. The intra-observer reliability was analyzed by the interclass correlation coefficient (ICC) test. The measurement error calculated using Dahlberg’s formula was 0.42 mm for linear measurements and 0.41° for angular measurements [14]. The ICCs of the selected variables between the first and second measurements indicated high reliability (>0.90).

## 3. Results

### 3.1. Patient Demography

Table 2 summarized the results for all patients included in this study. The two groups did not differ significantly with regard to demographic parameters.

### 3.2. Blood Loss and Operation Time

The mean operative time was 115.9 ± 22.5 min in the traditional RED group and 154.3 ± 24.8 min in the bone-borne RED group. The mean difference in surgical duration between the two groups was 38.4 min, and this was statistically significant (*p* < 0.005; Table 2).

The mean EBL was 154.6 ± 136.9 mL in the traditional RED group and 134.6 ± 90.7 mL in the bone-borne RED group, the amount of blood loss of bone-borne RED group was less than the traditional RED group by 20 mL, though this difference was not statistically significant (*p* = 0.80; Table 2).

### 3.3. Surgical Outcomes

The twenty-four variables were measured to assess the surgical outcomes in the bone-borne RED group and the traditional tooth-borne RED group. Twelve skeletal measurements (5 angular and 7 linear), eight dental measurements (2 angular and 6 linear) and four soft tissue linear measurements were used to analyze the changes between each stage. The pre-distraction parameters (T0) regarding skeletal, dental, and soft tissue elements between the bone-borne RED group and the traditional tooth-borne RED group revealed no significant difference (Table 3).

The serial cephalograms were taken at three different stages to determine the postdistraction result (T1), short-term (T2), and long-term (T3) stability within the both groups (Table 4 and Table 5), and the comparison of the bone-borne RED group and the traditional tooth-borne RED group were illustrated in Table 6.

### 3.4. Distraction Efficacy (T1–T0)

#### 3.4.1. Skeletal Changes

The mean increases of post-distraction ANB values in the bone-borne RED group and the traditional tooth-borne RED group were 19.45° and 14.51°, respectively. The post- distraction ANB changes demonstrated the significant improvement from Class III skeletal relationship to harmonious two-jaw relationship within each group (Table 6). The mean increases of SNA value were 19.72° in the bone-borne RED group and 15.01° in the traditional tooth-borne RED group, and the result has statistical significance. The similar changes were also found in the linear parameters. The linear changes of A-Nperp were 18.01 mm and 13.61 mm in the bone-borne RED group and the traditional tooth-borne RED group, respectively. As to the horizontal position changes of landmarks in ANS and A point, the bone-borne RED group showed a significantly greater maxillary advancement than traditional tooth-borne RED group after distraction: ANS (bone-borne RED group: 19.98 ± 5.64 mm; traditional tooth-borne RED group: 14.52 ± 4.62 mm) and A point (bone-borne RED group: 19.54 ± 5.61 mm; traditional tooth-borne RED group: 15.67 ± 4.95 mm).

In view of the changes in the orientation of the palatal plane related to the S-N plane (SN-PP), the traditional tooth-borne RED group illustrated significant counter-clockwise rotation in the palatal plane after distraction; whereas the bone-borne RED group presented with small degree of clockwise rotation (bone-borne RED group: 0.40° ± 4.17°; traditional tooth-borne RED group: −11.23° ± 4.90°). The rotational changes could also be confirmed from the linear parameters regarding the vertical changes of landmarks in ANS (bone-borne RED group: 2.29 ± 3.75 mm; traditional tooth-borne RED group: −4.29 ± 2.95 mm), PNS (bone-borne RED group: 0.90 ± 0.86 mm; traditional tooth-borne RED group: 4.97 ± 1.29 mm), and A point (bone-borne RED group: 2.40 ± 3.85 mm; traditional tooth-borne RED group: −4.20 ± 2.88 mm).

#### 3.4.2. Dental Changes

At the end of the distraction, the mean increases of overjet were 15.23 ± 3.66 mm in the bone-borne RED group and 16.97 ± 3.53 mm in the traditional tooth-borne RED group, and the upper incisors and molars showed great anterior movements within both groups (incisors: bone-borne RED group: 20.09 ± 6.59 mm; traditional RED group: 20.57 ± 5.59 mm; molars: bone-borne RED group: 18.66 ± 5.99 mm; traditional RED group: 20.84 ± 4.95 mm) (Table 6).

The results were found significantly different regarding the changes of upper incisal inclination related to the S-N plane (U1-SN°) and the palatal plane (U1-PP°). In the bone-borne RED group, the change of U1-PP° demonstrated a small degree of upper incisors’ retroclination after distraction, and the change of U1-SN° value showed a similar result (U1-PP°: −3.94° ± 10.66°; U1-SN°: −3.10° ± 9.19°), indicating a minimal dental side-effect and considerable translation movement along with skeletal advancement. However, the change of U1-PP ° exhibited a large degree of upper incisors’ retroclination in the traditional tooth-borne RED group, which was not comparable to that of the U1-SN ° value (U1-PP°: −10.86 ° ± 12.59°; U1-SN°: −0.95° ± 11.80°).

#### 3.4.3. Soft Tissue Changes

The soft tissue landmarks of soft tissue A point (A’) and pronasale showed significant forward and upward movements after maxillary distraction. The post-distraction positional changes of soft tissue A’ point in the bone-borne RED group moved anteriorly by 19.14 ± 5.65 mm and superiorly by 1.32 ± 2.70 mm, whereas in the traditional tooth-borne RED group were 15.57 ± 6.04 mm anteriorly and 3.06 ± 2.99 superiorly. Furthermore, the horizontal changes of pronasale were 9.17 ± 3.32 mm and 8.12 ± 4.26 mm in the bone-borne RED group and traditional tooth-borne RED group, respectively. The vertical position of pronasale moved more superiorly in the traditional tooth-borne RED group by 4.60 ± 2.60 mm than in the bone-borne RED group by 2.19 ± 2.43 mm, which resulted from the counter-clockwise rotation of the maxilla in the traditional tooth-borne RED group.

### 3.5. Shot-Term and Long-Term Stability

#### 3.5.1. Skeletal Stability

Short-term and long-term stability were evaluated at 6 months (T2) and 1.5 years (T3), respectively, after the cease of consolidation phase and the removal of the RED system. With regard to short-term stability, the maxilla tended to move backward with the decrease of SNA angle within the both groups (bone-borne RED group: −2.10° ± 2.06°, traditional tooth-borne RED group: −2.65° ± 2.32°), and the similar results were also found in the change of A-Nperp value and the horizontal changes in the ANS and the A point (Table 6). The backward movement of maxilla continued until T3; although the changes of SNA angle was greater in the bone-borne RED group (−1.61° ± 1.58°) than in the traditional tooth-borne RED group (−0.76° ± 1.43°), the result is of no significance (*p* = 0.223).

Regarding the changes of vertical position, the traditional tooth-borne RED group a showed greater amount of inferior movement in the ANS than the bone-borne RED group in T2 (bone-borne RED group: 0.69 ± 1.67 mm, traditional RED group: 2.54 ± 2.14 mm). The finding was in line with the changes of SN-PP value in T2, with more degrees of clockwise rotation in the traditional tooth-borne RED group than the bone-borne RED group after the 6-month post-distraction follow-up period (bone-borne RED group: 1.11° ± 1.28°, traditional tooth-borne RED group: 3.42° ± 2.08°).

#### 3.5.2. Relapse Rate

After 6 months (T2–T1), the ANS showed a horizontal backward relapse of −1.60 ± 1.38 mm in the bone-borne RED group and −2.89 ± 2.39 mm in the traditional tooth-borne RED group, and the relapse rate in the horizontal direction was 7.7% and 18.6%, respectively. The A point revealed a similar result of backward movement; the horizontal relapse rate was 1.71 ± 1.36 mm (7.9%) in the bone-borne RED group and 2.75 ± 2.25 mm (17.2%) in the traditional tooth-borne RED group.

After 1.5 years (T3–T1), the ANS showed a horizontal backward relapse of −2.55 ± 1.43 mm in the bone-borne RED group and −3.50 ± 2.49 mm in the traditional tooth-borne RED group, whereas the A point presented a horizontal relapse of −2.91 ± 1.84 mm in the bone-borne RED group and −3.50 ± 2.66 mm in the traditional tooth-borne RED group. The long-term horizontal relapse rate in ANS were 12.9% in the bone-borne RED group and 24.1% in the traditional tooth-borne RED group, and the A point showed long-term horizontal relapse rate by 14.9% and 22.3%, respectively (Table 7).

#### 3.5.3. Angular and Linear Change of Upper Central Incisors

The orthodontic treatment was initiated before distraction and was continued throughout T2 and T3 in the traditional tooth-borne RED group, whereas in the bone-borne RED group the start of orthodontic treatment was between T2 and T3. The overjet reduction occurred throughout short-term and long-term follow-up period, with greater amount of reduction in T2 and gradually decreased until T3 (Table 6). The values of U1-SN and U1-PP parameters increased in the traditional tooth-borne RED group in T2 and T3, indicating the proclination of upper incisors in long-term follow-up period. However, in the bone-borne RED group, the incisors exhibited retroclination in T2 and gradually became proclination until T3. The timing of orthodontic treatment intervention may result in the differences between the two groups.

#### 3.5.4. Soft Tissue Changes

The changes of the pronasale and the soft tissue A’ point showed backward and inferior relapse in short-term period, with more relapse rate in the traditional tooth-borne RED group than in the bone-borne RED group (Table 6). The amount of relapse gradually decreased over time in all the soft tissue parameters except the vertical position of soft tissue A’ point, which was affected by the soft tissue growth and upper incisal position.

## 4. Discussion

Cleft-related maxillary hypoplasia as a congenital craniofacial deformity tends to be managed with surgical repositioning of the maxilla to restore facial esthetics and function [15]. Correcting a developing skeletal discrepancy in childhood as soon as possible also tends to reduce psychological stress and enhance occlusal function. Maxillary DO is introduced as a novel alternative surgical technique to treat young CL/P patients with maxillary retrusion before the completion of facial growth, with no need to wait for bimaxillary orthognathic surgery after skeletal maturity [1,2,3,4]. Currently, external and internal distraction devices are two popular types of maxillary distraction system [4,16]. Of these two systems, the external distraction device is easier to equip by surgeons and to adjust by orthodontists. Traditionally, the tooth-borne RED system was introduced by connecting an intra-oral appliance onto the extra-oral distractor; however, patients with inadequate dentition cannot take the advantage of tooth-borne RED system due to difficulties in fixing the intra-oral appliance. The modification of the RED system by using bone-borne miniplates with transcutaneous wire instead of tooth-borne intra-oral appliance, as shown in this study, provides a simple, safe, and predictable treatment modality.

Maxillary DO with RED protocol involves four stages: LeFort I osteotomy for separating the maxilla from craniofacial structure, the latency stage of 3–5 days for callus formation, the distraction stage of 2–3 weeks for maxillary protraction, and the consolidation stage of 6–8 weeks for new bone maturation [2,6,17]. During the distraction stage, maxillary advancement is performed by turning the activating screw at a rate of 2 turns/1 mm per day [2]. The RED distractor gradually elongated the separated maxilla forward, and then new bone deposited and connected osteotomized bone edges. Rachmiel et al. found mature lamellar bone formation in the distraction site between two separated bony segments. The new mature lamellar bone provides a sound physical support to stabilize the advanced maxilla [18]. Kusnoto et al. demonstrated substantial bone formation in the pterygoid region after DO with RED, and the bone trabecular was seen on oriented tomography at 6 weeks after active distraction [19]. Figuerora et al. provided human histology evidence from a patient who had underwent the RED procedure and still underwent additional surgery afterward; as a result, they found well-ossified dense lamellar bone by obtaining a bone biopsy specimen at the pterygomaxillary region [2]. In addition, the gradual advancement of the maxilla allow soft tissue expansion and tissue regeneration [20]. The reliable biomechanics enable proper management of callus manipulation, progressive bone regeneration at pterygomaxillary region, and the induction of soft tissue adaption, which is termed distraction histogenesis and thought to be one of the merits of DO [21].

In the present study, both the bone-borne RED group and the traditional tooth-borne RED group exhibited the similar period of distraction under the same adjustment protocol as mentioned above. The horizontal changes in the ANS of the bone-borne RED group was advanced by 19.98 ± 5.64 mm after distraction; the result was in line with our previous literature conducted by Gao et al., who assess the effectiveness of maxillary DO by using bone-borne RED with a transcutaneous wire system to treat CL/P patients with severe maxillary hypoplasia, demonstrating considerable maxillary advancement by an average of 20.5 ± 5.1 mm [22]. In comparison to the traditional tooth-borne RED group, a greater amount of maxillary advancement in the bone-borne RED group was revealed in the mean increases of SNA angle by 4.71 degree, in ANS by 5.46 mm, in A point by 3.87 mm, and in A-Nperp by 4.4 mm. The bone-borne RED group displayed a higher efficacy of maxillary distraction than the traditional tooth-borne RED group, and the direction of distraction vector in relation to the center of resistance of the maxilla is thought to be the main reason for the difference. In the traditional tooth-borne RED group, the dramatic changes of post-distraction SN-PP angle decreased by −11.23°, indicating a counter-clockwise rotation of the maxilla during distraction. The maxilla moved upward considerably, thus reducing the amount of forward distraction. By contrast, the changes of post-distraction SN-PP angle in the bone-borne RED group increased by 0.4°, representing the purely anterior translation of maxillary component without undesired rotational side-effect during distraction. The distraction force directly passing through the center of resistance of the maxilla, which is located 5–10 mm below the orbitale on the zygomatic bone or at the apex of the maxillary premolars in the lateral view, is considered to be the desired distraction vector [4,23]. The traditional tooth-borne RED system incorporated an intra-oral splint with vertical projecting arms, and the height of arms enabled the orthodontist to control the distraction vector. However, the intra-oral splint was fabricated before surgery, and it was quite difficult to estimate the osteotomy line relative to the center of resistance of the maxilla in the pre-operative laboratory procedure. Besides, the construction of custom-made intra-oral appliance required a sophisticated wire bending technique, which contained errors in determining the vertical height of the arms. In addition, because of the inherent flexible characteristic of archwires, the vertical arms of the intra-oral splint deflected significantly under the application of distraction force during post-operative clinical adjustment. Consequently, the distraction vector may be far from that of the pre-planned desired direction in relation to the center of resistance of the maxilla, thus contributing to unwanted rotational side-effect during distraction. Unlike the traditional tooth-borne RED approach, the distraction force transmission directly linked to the fixed bone plates onto the maxilla through transcutaneous wires in the bone-borne RED system. Moreover, the position of bone plates could be fixed closer to the center of resistance of the maxilla in relation to the LeFort I osteotomy line during operation, providing an appropriate distraction vector for the horizontal translation of the maxilla without significant rotational side-effect. Therefore, these findings highlight the advantages of utilizing the bone-borne RED approach in treating young patients with cleft-related hypoplasia, thus facilitating the effectiveness of maxillary distraction.

Dental movements were examined by comparing the changes of linear and angular parameters of the maxillary incisors and first molars. The results of this study illustrated that dental components in both groups showed remarkable forward movement after distraction; however, there were significant differences between these two groups in dental movements relative to their skeletal component. In the bone-borne RED group, the mean amount of dental forward movement of incisors (20.09 ± 6.59 mm) was comparable with that of skeletal advancement in ANS (19.98 ± 5.64 mm). Maxillary teeth and their accompanying maxillary bone segment moved a similar range of distance, presenting a minimal side-effect over the dentition as the maxilla distracted forward, and the result was in accordance with our previous associated study (Gao et al.) [22]. On the contrary, the mean amount of dental forward movement of incisors was larger than skeletal advancement in ANS by 6.23 mm in the traditional tooth-borne RED group, indicating that dental movement exceeded bone movement through the traditional tooth-borne RED approach. The finding that greater anterior movement occurred in the upper incisors than in the ANS was in agreement with the previous RED studies conducted by Huang et al., Harada et al., and Aksu et al., which could be explained by the use of maxillary teeth as the anchor and the counter-clockwise rotation of maxilla after distraction [5,24,25].

In terms of the angular change of upper incisors, a varying degree of palatal inclination in relation to S-N plane and palatal plane was found between both groups. In the traditional tooth-borne RED group, the distracted maxilla moved in a counter-clockwise rotation pattern, and the angular changes of incisors were supposed to be increased in the U1-SN angle and unchanged in the U1-PP angle. However, the actual mean post-distraction changes of U1-PP presented with significant palatal inclination by −10.86 °, larger than that of U1-SN value of −0.95°. The increase of soft tissue tension could be the explanation for the huge differences. The greater extent of maxillary protraction generated stronger soft tissue tension immediately after distraction, thus imposing excessive stress on front teeth and giving rise to the palatal inclination of incisors. Suzuki et al. reported the similar findings of palatal inclination of incisors in 8 of 12 patients after distraction by using the traditional tooth-borne RED approach, whereas in our study these results were found in 9 of 11 individuals [26]. Unlike the traditional tooth-borne RED group, the angular changes of U1-SN by −3.10° was inconsistent with the changes of U1-PP by −3.94° in the bone-borne RED group, whereas the incisors’ inclination showed a little change in reference to both the S-N plane and the palatal plane. The results implied that the bone-borne RED directly distracted the maxilla by means of bone anchorage and transcutaneous wire passing through the lateral alar base, providing the forward translation the maxilla, better soft tissue adaption, and less skin tension than the tooth-borne RED approach. The bone-borne RED system is superior to traditional tooth-borne RED approach in reducing or avoiding increased soft tissue pressure after distraction, thus preventing the incisors from excessive palatal inclination and protecting the supporting periodontal tissue around the front teeth.

Overcorrection from the desired maxillary position before discontinuing activation of RED was advocated to accommodate for mandibular growth and to compensate for post-DO skeletal relapse. The distracted maxilla possessed limited growth ability due to either the obliteration of pterygomaxillary junction or soft tissue tension after extent great deal of distraction [5]. Therefore, the compensation for relapse after distraction is of crucial importance for preventing the recurrence of Class III malocclusion and establishing the final position of maxilla. In our study, the A point demonstrated greater advancement in the bone-borne RED group by 19.54 mm than the traditional tooth-borne RED group by 15.67 mm immediately after distraction. After the short-term period (T2-T1), the A point showed less mean horizontal relapse of −1.71 mm in the bone-borne RED group than the horizontal relapse of −2.75 mm exhibited by the traditional tooth-borne RED group. In the T2–T3 interval, the mean horizontal backward movement of the A point in bone-borne RED group gradually decreased to −1.20 mm, whereas the traditional tooth-borne RED group demonstrated lesser backward movement of −0.75 mm. The greater backward movements during T2–T3 in the bone-borne RED group may be attributable to the large extent of distraction distance. However, in reference to the post-distraction period, the short-term horizontal relapse rates (T2–T1) in the A point were 7.9% in the bone-borne RED group and 17.2% in the traditional tooth-borne RED group, whereas the long-term relapse rates (T3–T1) were 14.9% and 22.3%, respectively. As a consequence, the traditional tooth-borne RED approach showed a higher relapse rate than the bone-borne RED group in the long-term follow-up period. Currently, numerous studies have demonstrated the result of long-term follow-up by using traditional tooth-borne RED approach. Harada et al. reported a relapse rate of 12% (1.2 mm) after 10.1 mm advancement in a 36-months follow up [24]. Cho et al. conducted one- to six-year long-term assessment and revealed the relapse rate of 23% after 13.6 mm advancement; therefore, they claimed that an overcorrection of 20–30% is required for treating the patients with cleft-related maxillary hypoplasia [27]. In contrast, a greater relapse rate after the 3-year follow-up period was presented by Huang et al. (34% relapse after 9.4 mm advancement) and Aksu et al. (22% relapse after 9 mm advancement) [5,25]. Furthermore, Saltaji et al. summarized the finding of numerous studies and proposed that DO can be expected to relapse about 15% (1.5 mm) after 10 mm of the A point advancement [17]. A possible interpretation for the inconsistent outcomes could be attributed to the individual differences in the cleft structure and morphology between each study (i.e., difference in the side and extent of cleft defect, the severity of discontinuity maxillary segment, the scar contraction from lip, alveolus and palate). In comparison to previous traditional tooth-borne RED research, our study not only presented a greater distraction range than those reported studies, but also revealed the better outcome assessment of the innovative bone-borne RED system. Despite the fact that specific clinical guidelines regarding overcorrection in maxillary distraction have not addressed the compensation for the post-operative backward relapse of the maxilla, we suggested at least 15% of overcorrection for the bone-borne RED system and 25% for the traditional tooth-borne RED approach if the designed range of distraction was within the amount of 15 to 20 mm. This recommendation is based on the long-term relapse rate in the present study (the bone-borne group: −12.9% in the ANS(x) and −14.9% in the A-point; the tooth-borne group: −24.1% in the ANS(x) and −22.3% in the A-point).

The use of bone-borne RED with transcutaneous wire brings many benefits for treating patients with cleft-related maxillary hypoplasia. First, the transcutaneous wire is easier for orthodontist to directly control and to adjust the desired direction of distraction. Second, patients with inadequate dentition and minimal tooth anchor can take advantage of the bone-borne RED system. Lip irritation, chewing discomfort, and inadequate oral hygiene care could be eliminated without using intra-oral splint. Third, the bone-borne RED system did not require work for fabricating complicated intra-oral splint by dental laboratory technicians, which is extremely delicate and time-consuming. Although in this study, the operative time was lengthened by 38.4 min in the bone-borne RED group for fixing the miniplates, it did not seem to cause an inconvenience for the surgeon or increase intraoperative blood loss.

The complications showed in only 2 of 11 patients managed by bone-borne RED with transcutaneous wire system. One patient developed an infection over lateral alar base during the consolidation phase; the infected region was treated with proper medication therapy, and the process of distraction went smoothly and showed good results. Another patient experienced screw loosening before distraction; the screw was retightened under local anesthesia with outpatient procedure, and the screw remained stable throughout the distraction period. The fair-bone quality was detected by the surgeon during screw insertion; accordingly, bone quality could be one of the limitations with the use of bone-borne RED system. Other complications associated with the RED system, such as scalp pin loosening or headframe migration, were not shown in our study [28]. Nevertheless, neither complication interfered with the distraction procedure or required operative management, and the complications did not compromise the final outcomes.

The main limitation of this study was that the sample size used was quite small, and this could be attributed to the low prevalence of cleft deformities. Patients with CL/P are not common in Taiwan. According to Lei et al. and Chang et al., the incidence of patients with cleft lip and palate in Taiwan was 1.37 to 1.43 per 1000 births, and this restricted the sample size considerably [29,30]. Future studies incorporating larger sample sizes are necessary. However, we believe that the present study should be considered as a pilot study examining the feasibility of using the bone-borne RED system with skeletal anchorage for treating patient with cleft-related maxillary hypoplasia.

## 5. Conclusions

The result of our study demonstrated that maxillary distraction using the bone-borne rigid external device (RED) with skeletal anchorage and transcutaneous wire in treating young patient with cleft-related maxillary hypoplasia presented significant maxillary advancement, better outcomes, and reasonable relapse rate in the short-term and long-term period than the traditional tooth-borne RED with dental anchorage. In addition, the distraction vector is easier to control by using bone-borne RED with skeletal anchorage, and the unwanted rotational side-effect of maxilla can be eliminated during maxillary distraction. The inclination change of incisors is minimal, and the dental movements are comparable with that of bony movements. Overall, the present study highlights the advantages of utilizing the bone-borne RED system with skeletal anchorage in treating young patient with cleft-related maxillary hypoplasia.

## Figures and Tables

**Figure 1 jpm-12-01062-f001:**
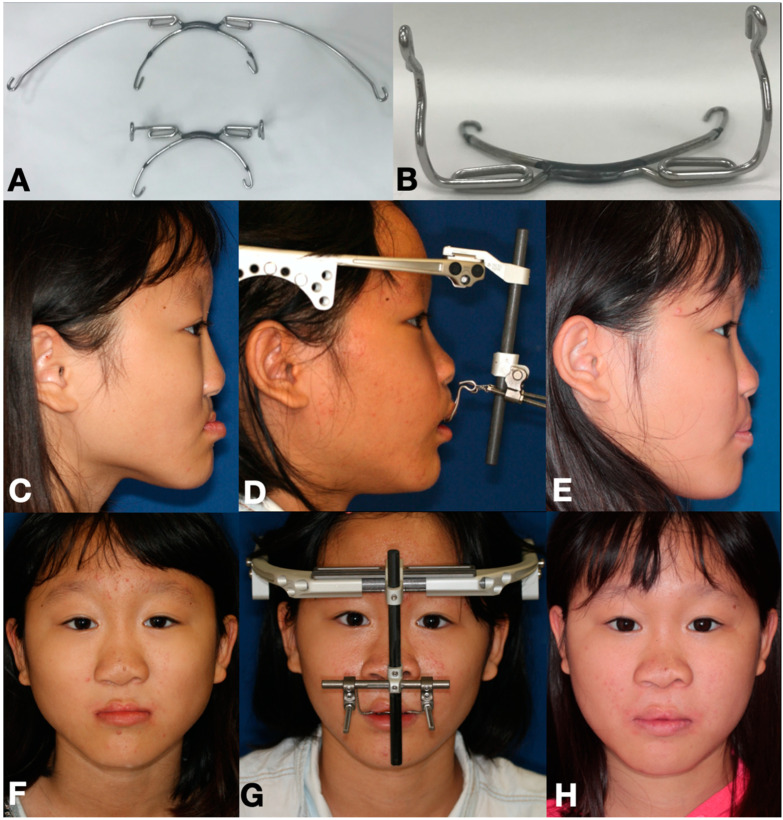
An eleven-year-old girl with cleft-related maxillary hypoplasia was treated with the traditional tooth-borne RED system for distraction osteogenesis. (**A**,**B**) Intra-oral splint was modified from orthodontic headgear appliance. (**C**–**E**) Lateral profile changes before surgery, immediate after distraction, and after 1.5-year review. (**F**–**H**) Frontal appearance changes during the three stages mentioned above.

**Figure 2 jpm-12-01062-f002:**
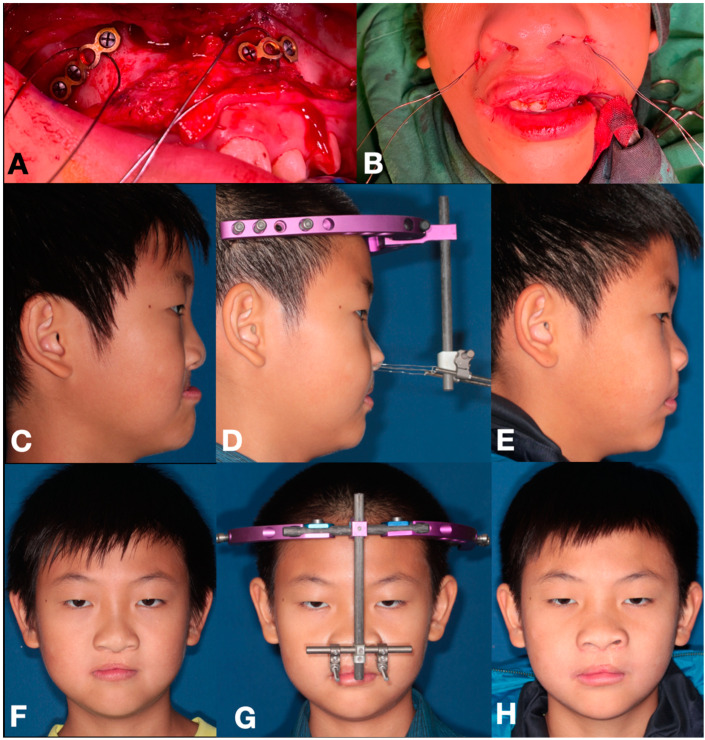
An eleven-year-old boy with cleft-related maxillary hypoplasia was treated with the bone-borne RED system for distraction osteogenesis. (**A**,**B**) Two miniplates were fixed with five bone screws on the osteotomized maxillary bony surface. Transcutaneous wires were ligated into the two un-screwed holes of the miniplates below LeFort I osteotomy line, and the wires were extended from the lateral sides of alar base. (**C**–**E**) Lateral profile changes before surgery, immediate after distraction, and after 1.5-year review. (**F**–**H**) Frontal appearance changes during the three stages mentioned above.

**Figure 3 jpm-12-01062-f003:**
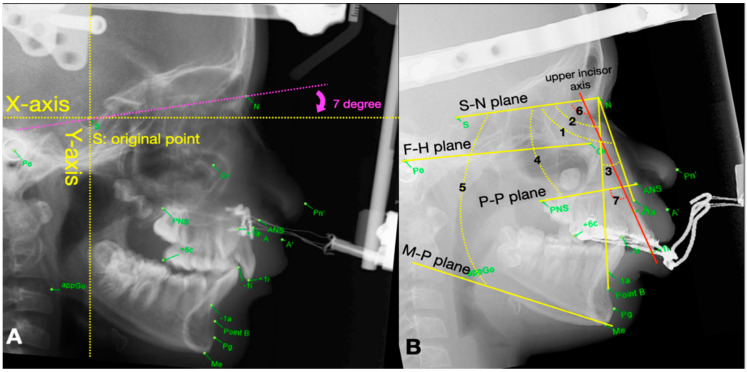
The lateral cephalograms of (**A**) Bone-borne RED with transcutaneous wire system present with cephalometric landmarks in relation to the X-Y coordinate system (**B**) traditional Tooth-borne RED with intra-oral splint, present with reference lines: S-N plane (S-N, sella-nasion plane); F-H plane (Po-Or, Frankfort plane); P-P plane (PNS-ANS, palatal plane); M-P plane (Go-Me, mandibular plane); upper incisor axis (+1i-+1a), and skeletal angular variables: 1, SNA; 2, SNB; 3, ANB; 4, SN-PP; 5, SN-MP, and dental angular variables: 6, U1-SN (upper incisor axis to S-N); 7, U1-PP (upper incisor axis to palatal plane) that selected in this study.

**Table 1 jpm-12-01062-t001:** Definition of the distance from the landmarks to the X-, *Y*-axis.

Parameters	Abbreviation	Definition
Linear parameters		
ANS to *X*-axis, mm	ANS (x), mm	The distance between the ANS to the *X*-axis. Points inferior to the *X*-axis are given a positive value.
ANS to *Y*-axis, mm	ANS (y), mm	The distance between the ANS to the *Y*-axis. Points anterior to the *Y*-axis are given a positive value.
PNS to *X*-axis, mm	PNS (x), mm	The distance between the PNS to the *X*-axis. Points inferior to the *X*-axis are given a positive value.
PNS to *Y*-axis, mm	PNS (y), mm	The distance between the PNS to the *Y*-axis. Points anterior to the *Y*-axis are given a positive value.
A point to *X*-axis, mm	A(x), mm	The distance between the A point to the *X*-axis. Points inferior to the *X*-axis are given a positive value.
A point to *Y*-axis, mm	A(y), mm	The distance between the A point to the *Y*-axis. Points anterior to the *Y*-axis are given a positive value.
Upper incisor to *X*-axis, mm	U1 (x), mm	The distance between the central incisor crown tip to the *X*-axis. Points inferior to the *X*-axis are given a positive value.
Upper incisor to *Y*-axis, mm	U1 (y), mm	The distance between the central incisor crown tip to the *Y*-axis. Points anterior to the *Y*-axis are given a positive value.
Upper molar to *X*-axis, mm	U6 (x), mm	The distance between the distal marginal ridge of first molar to the *X*-axis. Points inferior to the *X*-axis are given a positive value.
Upper molar to *Y*-axis, mm	U6 (y), mm	The distance between the distal marginal ridge of first molar to the *Y*-axis. Points anterior to the *Y*-axis are given a positive value.
Pronasale to *X*-axis, mm	Prn (x), mm	The distance between the nasal tip to the *X*-axis. Points inferior to the *X*-axis are given a positive value.
Pronasale to *Y*-axis, mm	Prn (y), mm	The distance between the nasal tip to the *Y*-axis. Points anterior to the *Y*-axis are given a positive value.
A’ soft tissue point to *X*-axis, mm	A’ (x), mm	T he distance between the soft tissue A point to the *X*-axis. Points inferior to the *X*-axis are given a positive value.
A’ soft tissue point to *Y*-axis, mm	A’ (y), mm	The distance between the soft tissue A point to the *Y*-axis. Points anterior to the *Y*-axis are given a positive value.
A point to Nperp	A-Nv	The distance between the A point to the Nperp reference line. Points anterior to the Nperp are given a positive value, while posterior to Nperp are assigned a negative value
Reference lines		
Frankfort horizontal plane	FH plane	Plane constructed by Po and Or
Nasion perpendicular line	Nperp	Nv Vertical reference to FH plane and passing through Nasion.

**Table 2 jpm-12-01062-t002:** Characteristics of the CL/P patients in the bone-borne RED group and the traditional tooth-borne RED group.

Variables	Bone-Borne	Tooth-Borne	*p* ^†^
Patients (*n*)	11	11	-
Gender (male/female)	7:4	8:3	-
Age (years)	11.5 ± 0.7	11.8 ± 0.5	0.141
Operative time (min)	154.3 ± 24.8	115.9 ± 22.5	<0.002 **
Estimated blood loss (mL)	134.6 ± 90.7	154.6 ± 136.9	0.808
Period of distraction (days)			
Latency stage	8.55 ± 3.05	8.09 ± 0.83	-
Distraction stage	31.1 ± 9.9	35.6 ± 11.5	-
Consolidation stage	69.0 ± 11.5	72.9 ± 20.9	-

Data are presented as mean ± SD. Abbreviations: CL/P, cleft lip and palate; RED, rigid external device; *n*, number; min, minutes; ml, milliliter; SD, standard deviation. ^†^ Student’s *t*-test, significant level: ** *p* < 0.005.

**Table 3 jpm-12-01062-t003:** The pre-distraction cephalometric measurements in the bone-borne RED group and the traditional tooth-borne RED group.

Variables	Bone-Borne	Tooth-Borne	*p* ^†^
Skeletal			
SNA°	75.56 ± 3.12	74.50 ± 3.26	0.401
SNB°	80.83 ± 4.49	79.96 ± 2.70	0.748
ANB°	−5.28 ± 4.88	−5.50 ± 2.70	0.606
SN-MP°	33.60 ± 5.36	32.73 ± 4.75	0.562
SN-PP°	9.59 ± 5.62	11.91 ± 4.29	0.438
ANS (x), mm	57.33 ± 3.04	60.11 ± 6.77	0.478
ANS(y), mm	42.40 ± 2.53	45.20 ± 4.16	0.088
PNS (x), mm	17.68 ± 4.16	17.57 ± 3.10	0.943
PNS (y), mm	40.90 ± 3.48	40.91 ± 4.13	0.996
A(x), mm	55.07 ± 2.49	57.23 ± 6.63	0.748
A(y), mm	46.91 ± 3.57	48.93 ± 4.39	0.270
A-Nv, mm	−6.68 ± 4.48	−7.59 ± 5.42	0.478
Dental			
Overjet, mm	−9.31 ± 3.82	−7.69 ± 3.76	0.365
Overbite, mm	1.62 ± 4.18	5.54 ± 2.59	0.065
U1-SN°	93.76 ± 11.34	93.42 ± 5.55	1.000
U1-PP°	103.68 ± 11.06	105.29 ± 6.50	0.652
U1 (x), mm	53.99 ± 3.63	57.27 ± 7.05	0.478
U1 (y), mm	64.89 ± 5.51	70.67 ± 7.04	0.076
U6 (x), mm	22.16 ± 3.28	24.07 ± 3.95	0.365
U6 (y), mm	59.66 ± 5.87	62.70 ± 5.99	0.401
Soft tissue			
Prn (x), mm	81.12 ± 3.82	85.09 ± 7.62	0.270
Prn (y), mm	34.71 ± 4.26	40.19 ± 4.91	0.143
A’ (x), mm	68.79 ± 3.94	72.60 ± 7.82	0.217
A’ (y), mm	52.94 ± 5.23	57.77 ± 5.49	0.121

Data are presented as mean ± SD. Abbreviations: (x), horizontal change; (y), vertical change; RED, rigid external device; SD, standard deviation. ^†^ Student’s *t*-test.

**Table 4 jpm-12-01062-t004:** The mean changes of the skeletal, dental and soft tissue components based on serial cephalograms in the traditional tooth-borne RED group.

Variables	T_0_	T_1_	T_2_	T_3_	*p* ^†^		
					T1–T0	T2–T1	T3–T2
Skeletal							
SNA°	74.50 ± 3.26	89.97 ± 4.31	87.32 ± 3.62	86.56 ± 3.56	***	*	0.248
SNB°	79.87 ± 2.85	80.37 ± 3.90	80.63 ± 3.68	81.01 ± 3.92	0.328	0.350	0.453
ANB°	−5.50 ± 2.70	9.38 ± 4.34	6.72 ± 3.67	5.46 ± 3.41	***	**	*
SN-MP°	32.33 ± 4.75	33.88 ± 4.26	33.61 ± 3.62	33.28 ± 3.92	0.213	0.790	0.477
SN-PP°	11.91 ± 4.29	0.67 ± 4.93	4.09 ± 4.28	5.16 ± 4.41	***	**	0.073
ANS (x), mm	60.11 ± 6.77	74.34 ± 5.48	72.17 ± 5.64	71.56 ± 5.83	***	**	0.215
ANS(y), mm	45.20 ± 4.16	40.91 ± 4.47	43.45 ± 4.38	45.23 ± 4.61	**	*	*
PNS (x), mm	17.57 ± 3.10	31.50 ± 5.52	28.65 ± 4.45	28.07 ± 4.97	***	**	0.477
PNS (y), mm	40.91 ± 4.13	46.49 ± 4.93	45.93 ± 4.32	46.73 ± 4.19	**	0.424	*
A(x), mm	57.23 ± 6.63	72.09 ± 5.09	70.14 ± 5.12	69.39 ± 5.55	***	**	0.657
A(y), mm	48.93 ± 4.39	44.73 ± 4.53	46.76 ± 4.32	48.99 ± 4.57	**	*	*
A-Nv, mm	−7.59 ± 5.42	6.02 ± 4.24	4.16 ± 4.22	3.62 ± 4.25	***	*	0.262
Dental							
Overjet, mm	−7.59 ± 3.42	9.38 ± 3.45	5.61 ± 2.98	4.49 ± 2.36	***	**	0.097
Overbite, mm	5.02 ± 2.85	5.24 ± 3.51	2.90 ± 2.33	2.04 ± 2.52	0.722	*	0.110
U1-SN°	93.42 ± 5.55	92.47 ± 10.13	91.90 ± 15.07	98.82 ± 13.38	0.795	0.722	*
U1-PP°	105.29 ± 6.50	94.43 ± 12.17	95.67 ± 14.02	101.91 ± 11.28	*	0.424	*
U1 (x), mm	70.67 ± 7.04	74.72 ± 9.13	74.79 ± 7.93	75.23 ± 7.91	*	0.790	0.477
U1 (y), mm	57.27 ± 7.05	77.66 ± 8.67	74.32 ± 8.20	74.78 ± 8.86	**	**	0.505
U6 (x), mm	62.70 ± 5.99	65.23 ± 9.94	67.13 ± 6.84	67.92 ± 7.07	0.274	0.286	0.213
U6 (y), mm	24.07 ± 3.95	44.62 ± 4.61	41.72 ± 4.62	40.29 ± 5.64	**	**	0.062
Soft Tissue							
Prn (x), mm	85.09 ± 7.62	92.61 ± 8.73	91.04 ± 9.28	93.76 ± 8.03	**	*	*
Prn (y), mm	40.19 ± 4.91	35.59 ± 5.40	38.11 ± 6.37	39.97 ± 5.51	**	**	0.091
A’ (x), mm	72.60 ± 7.82	87.88 ± 8.03	84.61 ± 9.58	85.70 ± 8.94	**	**	0.722
A’ (y), mm	57.77 ± 5.49	54.71 ± 4.80	56.59 ± 5.39	58.30 ± 5.95	*	*	*

Data are presented as mean ± SD. Abbreviations: (x), horizontal change; (y), vertical change; RED, rigid external device; SD, standard deviation; T0, before distraction; T1, 1-month after distraction; T2, 6 months after distraction; T3, 1.5 years after distraction. ^†^ Paired Sample *t*-test, significant level: * *p* < 0.05, ** *p* < 0.005, *** *p* < 0.001.

**Table 5 jpm-12-01062-t005:** The mean changes of the skeletal, dental, and soft tissue components based on serial cephalograms in the bone-borne RED group.

Variables	T_0_	T_1_	T_2_	T_3_	*p* ^†^		
					T1–T0	T2–T1	T3–T2
Skeletal							
SNA°	75.56 ± 3.12	94.66 ± 7.02	92.23 ± 7.35	90.62 ± 6.63	**	*	*
SNB°	81.14 ± 4.99	81.41 ± 5.07	82.52 ± 4.98	82.07 ± 5.05	0.722	*	0.182
ANB°	−5.28 ± 4.88	12.25 ± 4.02	9.73 ± 3.41	8.80 ± 3.40	***	*	*
SN-MP°	33.60 ± 5.36	33.30 ± 5.47	33.15 ± 5.21	34.09 ± 5.20	0.449	0.541	0.386
SN-PP°	9.19 ± 5.62	9.59 ± 3.32	9.97 ± 3.71	10.58 ± 4.03	0.790	*	0.721
ANS (x), mm	57.33 ± 3.04	76.60 ± 4.89	75.18 ± 4.74	74.23 ± 4.78	***	**	*
ANS(y), mm	42.40 ± 2.53	44.70 ± 5.17	45.44 ± 5.66	47.24 ± 5.50	0.091	0.241	*
PNS (x), mm	17.68 ± 4.16	34.22 ± 6.51	32.68 ± 7.24	30.49 ± 6.90	***	*	*
PNS (y), mm	40.90 ± 3.48	41.81 ± 3.40	42.26 ± 3.95	43.25 ± 3.80	**	0.646	**
A(x), mm	55.07 ± 2.49	74.61 ± 5.08	72.94 ± 5.24	71.73 ± 5.11	***	**	**
A(y), mm	46.91 ± 3.57	49.31 ± 6.60	50.57 ± 7.27	52.07 ± 7.08	0.075	0.139	*
A-Nv, mm	−6.68 ± 4.48	11.33 ± 6.05	9.05 ± 6.37	7.69 ± 5.71	**	*	*
Dental							
Overjet, mm	−9.55 ± 3.61	5.68 ± 3.07	2.94 ± 2.18	2.10 ± 2.39	***	**	0.195
Overbite, mm	1.75 ± 3.79	1.62 ± 3.56	2.17 ± 2.65	0.60 ± 2.14	0.657	0.169	*
U1-SN°	93.76 ± 11.34	90.66 ± 7.71	90.19 ± 5.25	91.38 ± 11.08	0.110	0.333	0.203
U1-PP°	103.68 ± 11.06	99.75 ± 6.90	100.12 ± 5.33	102.38 ± 10.39	0.249	0.646	0.093
U1 (x), mm	53.99 ± 3.63	73.25 ± 5.41	71.98 ± 5.18	70.41 ± 5.10	**	*	0.241
U1 (y), mm	64.89 ± 5.51	67.89 ± 8.59	69.48 ± 9.03	65.47 ± 22.33	*	*	0.139
U6 (x), mm	22.16 ± 3.28	40.82 ± 4.86	39.40 ± 4.08	37.95 ± 3.58	**	**	*
U6 (y), mm	59.66 ± 5.87	62.13 ± 6.79	62.00 ± 6.60	64.88 ± 8.00	*	0.386	**
Soft Tissue							
Prn (x), mm	81.12 ± 3.82	89.68 ± 3.35	89.52 ± 2.94	90.04 ± 3.34	**	0.646	*
Prn (y), mm	34.71 ± 4.26	32.51 ± 5.68	33.93 ± 6.84	35.22 ± 6.45	*	0.285	*
A’ (x), mm	68.79 ± 3.94	86.61 ± 4.46	83.98 ± 3.74	83.00 ± 3.36	**	**	*
A’ (y), mm	52.94 ± 5.23	51.62 ± 6.37	52.68 ± 7.55	54.69 ± 6.98	0.131	0.386	*

Data are presented as mean ± SD. Abbreviations: (x), horizontal change; (y), vertical change; RED, rigid external device; SD, standard deviation; T0, before distraction; T1, 1-month after distraction; T2, 6 months after distraction; T3, 1.5 years after distraction. ^†^ Paired Sample *t*-test, significant level: * *p* < 0.05, ** *p* < 0.005, *** *p* < 0.001.

**Table 6 jpm-12-01062-t006:** The comparison of mean changes regarding skeletal, dental, and soft tissue components between the bone-borne RED and the traditional tooth-borne RED group.

	△T_1_ (T_1_–T_0_)			△T_2_ (T_2_–T_1_)			△T_3_ (T_3_–T_2_)		
Variables	Bone-Borne	Tooth-Borne	*p* ^†^	Bone-Borne	Tooth-Borne	*p* ^†^	Bone-Borne	Tooth-Borne	*p* ^†^
Skeletal									
SNA°	19.72 ± 6.21	15.01 ± 3.17	*	−2.10 ± 2.06	−2.65 ± 2.32	0.426	−1.61 ± 1.58	−0.76 ± 1.43	0.223
SNB°	0.27 ± 1.60	0.49 ± 1.66	0.797	1.22 ± 1.71	0.26 ± 1.03	0.173	−0.49 ± 1.02	0.39 ± 1.65	0.114
ANB°	19.45 ± 4.82	14.51 ± 3.02	0.063	−3.32 ± 2.10	−2.91 ± 1.86	0.557	−1.12 ± 1.07	−1.15 ± 1.56	0.918
SN-MP°	−0.30 ± 1.28	0.75 ± 1.69	0.151	0.20 ± 1.08	0.13 ± 1.45	1.000	0.62 ± 1.79	−0.33 ± 1.53	0.197
SN-PP°	0.40 ± 4.17	−11.23 ± 4.90	***	1.11 ± 1.28	3.42 ± 2.08	*	0.29 ± 1.61	1.07 ± 1.77	0.251
ANS (x), mm	19.98 ± 5.64	14.52 ± 4.62	*	−1.60 ± 1.38	−2.89 ± 2.39	0.061	−0.95 ± 1.17	−0.61 ± 1.80	*
ANS(y), mm	2.29 ± 3.75	−4.29 ± 2.95	***	0.69 ± 1.67	2.54 ± 2.14	*	1.80 ± 1.24	1.78 ± 1.76	0.973
PNS (x), mm	16.54 ± 6.31	13.93 ± 5.40	0.309	−1.78 ± 2.24	−2.84 ±2.02	0.268	−1.79 ± 1.87	−1.58 ± 2.94	0.280
PNS (y), mm	0.90 ± 0.86	4.97 ± 1.29	***	−0.04 ± 1.76	−0.56 ±1.89	0.523	1.45 ± 1.38	0.80 ±0.76	0.251
A(x), mm	19.54 ± 5.61	15.67 ± 4.95	0.171	−1.71 ± 1.36	−2.75 ± 2.25	0.085	−1.20 ± 0.69	−0.75 ± 1.53	*
A(y), mm	2.40 ± 3.85	−4.20 ± 2.88	***	0.92 ± 1.65	2.03 ± 2.39	0.349	1.50 ± 1.29	2.23 ± 1.98	0.349
A-Nv, mm	18.01 ± 3.64	13.61 ± 4.07	*	−1.98 ± 1.85	−1.86 ± 3.29	0.863	−1.36 ± 1.52	−0.53 ± 1.28	0.173
Dental									
Overjet, mm	15.23 ± 3.66	16.97 ± 3.53	0.171	−3.17 ± 1.86	−3.77 ± 2.30	0.512	−0.84 ± 1.90	−1.11 ± 2.03	0.705
Overbite, mm	−0.12 ± 1.59	0.22 ± 4.68	0.748	0.60 ± 1.91	−2.33 ± 1.79	0.152	−1.69 ± 1.35	−0.86 ± 1.31	0.173
U1-SN°	−3.10 ± 9.19	−0.95 ± 11.80	0.438	−1.51 ± 4.58	2.24 ± 6.53	0.218	1.04 ± 9.79	6.92 ± 8.03	0.152
U1-PP°	−3.94 ± 10.66	−10.86 ± 12.59	0.401	−0.33 ± 3.98	3.91 ± 9.43	0.190	1.33 ± 8.48	6.24 ± 7.06	0.387
U1 (x), mm	20.09 ± 6.59	20.57 ± 5.59	0.606	−1.70 ± 1.93	−3.71 ± 2.15	*	−1.16 ± 3.25	0.46 ± 2.47	0.468
U1 (y), mm	3.01 ± 3.79	4.05 ± 5.40	0.478	1.10 ± 1.69	0.06 ± 2.70	0.173	2.02 ± 1.50	0.44 ± 1.55	*
U6 (x), mm	18.66 ± 5.99	20.84 ± 4.95	0.401	−2.06 ± 1.53	−3.18 ± 1.99	0.197	−1.46 ± 1.54	−1.43 ± 2.19	0.863
U6 (y), mm	2.48 ± 2.86	2.53 ± 7.24	0.193	−0.58 ± 2.13	1.90 ± 4.94	0.251	2.87 ± 2.09	0.80 ± 1.79	*
Soft tissue									
Prn (x), mm	9.17 ± 3.32	8.12 ± 4.26	0.684	−0.24 ± 1.09	−2.42 ± 2.88	*	0.52 ± 0.72	2.72 ± 3.31	0.223
Prn (y), mm	−2.19 ± 2.43	−4.60 ± 2.60	*	1.31 ± 2.85	2.36 ± 2.14	0.387	1.29 ± 1.50	1.37 ± 2.40	0.654
A’ (x), mm	19.14 ± 5.65	15.57 ± 6.04	0.247	−2.60 ± 1.58	−3.78 ± 3.69	0.756	−0.78 ± 0.81	−0.19 ± 2.65	0.796
A’ (y), mm	−1.32 ± 2.70	−3.06 ± 2.99	0.193	0.81 ± 2.44	1.88 ± 2.53	0.349	2.01 ± 1.57	1.71 ± 1.78	0.557

Data are presented as mean ± SD. Abbreviations: (x), horizontal change; (y), vertical change; RED, rigid external device; SD, standard deviation; T0, before distraction; T1, 1-month after distraction; T2, 6 months after distraction; T3, 1.5 years after distraction. ^†^ Independent Sample *t*-test, significant level: * *p* < 0.05, *** *p* < 0.001.

**Table 7 jpm-12-01062-t007:** The horizontal relapse rate in ANS(x) and A(x) in short-term and long-term follow-up.

	Short-Term Period		Long-Term Period	
Variables	Bone-Borne	Tooth-Borne	Bone-Borne	Tooth-Borne
ANS(x)	−7.7%	−18.6%	−12.9%	−24.1%
A(x)	−7.9%	−17.2%	−14.9%	−22.3%

Data are presented as %. Abbreviations: (x), horizontal change; short-term period, 6 months follow-up, short-term relapse rate was measured as (T2–T1)/(T1–T0); long-term period, 1.5 years follow-up, long-term relapse rate was measured as (T3–T1)/(T1–T0).

## Data Availability

All data generated during the study are presented in this paper.

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
