# Peer review of "A Comparative Study of Skeletal and Dental Outcome between Transcutaneous External Maxillary Distraction Osteogenesis and Conventional Rigid External Device in Treating Cleft Lip and Palate Patients"

_jpm, 2022, doi:10.3390/jpm12071062_

Round 1

Reviewer 1 Report

The authors described "A Comparative Study of Skeletal and Dental Outcome between Transcutaneous External Maxillary Distraction Osteogenesis and Conventional Rigid External Device in Treating Cleft Lip and Palate Patients". As they pointed out, no studies have reported the maxillary distraction efficacy of innovative bone-borne RED system with skeletal anchorage in comparison with traditional tooth-borne RED approach with dental anchorage. So, this study should be informative and attractive for potential readers. I have some question.

1. A schema describing the site of two anchoring miniplates should be added. Also, how many screws were used ?

2.  In Discussion, overcorrection from the desired maxillary position before discontinuing activation of RED is crucial in this surgery. Despite specific clinical guideline regarding overcorrection after distraction have not yet been laid to compensate for relapse, they suggested at least 15% of overcorrection for bone-borne RED system and 25% for traditional tooth-borne RED approach under the circumstance of 15 to 20 mm distraction distance. Why did the authors suggest 15 % overcorrection? Please add more details based on this study.

Author Response

JPM-1791255

Title: A Comparative Study of Skeletal and Dental Outcome between Transcutaneous External Maxillary Distraction Osteogenesis and Conventional Rigid External Device in Treating Cleft Lip and Palate Patients

 Response to Reviewers’ Comments: We are grateful for the reviewers' excellent suggestions, which will improve the quality and clarity of our manuscript. We have responded to the comments or questions and made the necessary changes in the revised manuscript. These portions are all highlighted in yellow color.

 Response to Reviewer 1 Comments

  1. A schema describing the site of two anchoring miniplates should be added. Also, how many screws were used?

 Response: The number of bone screws used to fix miniplates depends on the attempt to achieve the desired stability of the miniplates on the osteotomized maxilla during distraction. Since the patients of the bone-borne group all had osteotomy, the stability of the miniplate against distraction force is not critical. The issue would be the size of the miniplates, which would be associated with the number of the screw holes. Miniplates need to fit the bony contour, and the anatomic structures would limit the shape and size of miniplates. However, one screw hole of each miniplate has to be used for applying the transcutaneous wire. We have revised the Material and Methods (in session 2.3. Surgical technique) and part of the text for Figure 2. We are grateful for the comment from the reviewer.

Below please find the changed parts of the revised manuscript:

Materials and Methods
2.3. Surgical technique

Page 4
Line 164-166

Following exposure of the maxillary bone surface, the bone-borne RED group incorporated two anchoring miniplates that were inserted bilaterally onto the medial buttresses of the mobilized maxilla. A total of five bone screws were used to stabilize the two miniplates, and one screw hole of each miniplate was left for applying the transcutaneous stainless-steel wires, and subsequently, the wire passed out through the lateral alar base via skin incision. The intra-oral incisions were closed afterward (Figure 2 A, B).

 Figure 2.

Page 5

Line 172-175

An eleven-year-old boy with cleft-related maxillary hypoplasia was treated with the bone-borne RED system for distraction osteogenesis. (A, B) Two miniplates were fixed with five bone screws on the osteotomized maxillary bony surface. Transcutaneous wires were ligated into the two un-screwed holes of the miniplates below LeFort I osteotomy line, and the wires were extended from the lateral sides of alar base.

  1. In Discussion, overcorrection from the desired maxillary position before discontinuing activation of RED is crucial in this surgery. Despite specific clinical guideline regarding overcorrection after distraction have not yet been laid to compensate for relapse, they suggested at least 15% of overcorrection for bone-borne RED system and 25% for traditional tooth-borne RED approach under the circumstance of 15 to 20 mm distraction distance. Why did the authors suggest 15 % overcorrection? Please add more details based on this study.

Response:  We appreciate the reviewer for the comment. The suggestion of  “at least 15% of overcorrection for bone-borne RED system and 25% for traditional tooth-borne RED approach under the circumstance of 15 to 20 mm distraction distance” was based on our result of 1.5-year long-term relapse rates  (the bone-borne group: -12.9% in the ANS(x) and -14.9% in the A-point; the tooth-borne group: -24.1% in the ANS(x) and -22.3% in the A-point) that indicated in the Table 7. Numerous studies with the traditional tooth-borne RED approach have recommended the amount of overcorrection according to their long-term relapse rate. The study of the tooth-borne RED system by Cho et al. revealed the reduction was from an average of 13.6 mm (range: 10 to 15 mm) to 10.8 mm in 6 months and 10.4 mm in 1-6 years. The long-term relapse rates were 20.7% in 6 months and 23.0% in 1 to 6 years. Therefore, they recommended 20 to 30 percent overcorrection in maxillary advancement to compensate for the long-term post-operative maxillary backward relapse. (please refer to Reference 30: Cho, B.C.; Kyung, H.M. Distraction Osteogenesis of the Hypoplastic Midface using a Rigid External Distraction System: The Results of a One- to Six-Year Follow-Up. Plast Reconstr Surg. 2006, 118, 1201–1212.). Moreover, the long-term relapse rate of the bone-borne RED system has not been documented in the literature. Hence, we believed our results of long-term relapse rate in tooth-born RED and bone-borne RED in the present study could provide informative data for determining the amount of overcorrection.

In order to improve the quality and clarity of the manuscript, we have revised the Discussion parts according to the comment of reviewer. We are grateful for the helpful comment from the reviewer.

Below please find the changed parts of the revised manuscript:

Discussion
Page 16
Line 612-619

Despite the fact that specific clinical guidelines regarding overcorrection in maxillary distraction have not addressed the compensation for post-operative backward relapse of the maxilla, we suggested at least 15% of overcorrection for the bone-borne RED system and 25% for the traditional tooth-borne RED approach if the designed range of distraction is within the amount of 15 to 20 mm. This recommendation is based on the long-term relapse rate in the present study (the bone-borne group: -12.9% in the ANS(x) and -14.9% in the A-point; the tooth-borne group: -24.1% in the ANS(x) and -22.3% in the A-point).

Reviewer 2 Report

Dear authors,

This manuscript is interesting, especially for orthodontists.

However, in the orthodontic world, the superiority of the bone bourne expander compared to the tooth bourne is well known. What is the novelty of this manuscript?

Did you perform a NNT (number needed to treat) or a intention to treat?

Line – should be 3.3. S

Were the evaluators blindeded?

References should be in journal style and updated, there are several references too old: ex  1992, 1997, 1999, 2001,1993

Please define the following in the manuscript:

ANS (x), mm

ANS(y), mm

PNS (x), mm

PNS (y), mm

A(x), mm

A(y), mm

A-Nv, mm

U1 (x), mm

U1 (y), mm

U6 (x), mm

U6 (y), mm

Prn (x), mm

Prn (y), mm

A_’ _(_x_)_,_ _m_m_ _

A_’ _(_y_)_,_ _m_m_ _

Kind regards,

Author Response

Response to Reviewer 2 Comments

  1. In the orthodontic world, the superiority of the bone borne expander compared to the tooth borne is well known. What is the novelty of this manuscript?

Response: We thank the reviewer for this comment. The novelty of this manuscript is to provide informative data in correcting anteroposterior maxillary discrepancy by bone-borne and tooth-borne distraction methods in managing patients with cleft-related maxillary hypoplasia. The application of an orthodontic rapid maxillary expander is to manage the narrow dental arch (or palatal constriction) and correct an insufficiently developed maxilla in the transverse dimension. The advantage of the bone-borne palatal expander in achieving more orthopedic transverse expansion compared to the tooth-borne expander is well set up. However, in the present study, the bone-borne rigid external device (bone-borne RED) is to correct the anteroposterior maxillary discrepancy in those patients with significant cleft-related maxillary hypoplasia. The forward distraction/advancement of the maxilla is not to expand the dental arch in the transverse direction but to position the maxilla in a harmonic relationship with the frontal head and the mandible and to expand the relevant soft tissue envelope over the face and jaw. No studies have reported the maxillary distraction efficacy of this innovative bone-borne RED system with skeletal anchorage compared with the traditional tooth-borne RED approach with dental anchorage.

  1. Did you perform a NNT (number needed to treat) or a intention to treat?

Response: We thank the reviewer for this comment. Please find below point-by-point responses to the reviewer’s comments.

  1. The Number Needed to Treat (NNT) is to measure the treatment effectiveness by evaluating the number of patients who require treatment. This measure is also to avoid the potential improper outcome. The NNT is performed by comparing the number of patients that need treatment to benefit for one from a control in a clinical trial and is calculated as the inverse of the absolute risk reduction (ARR), i.e., the difference between the experimental event rate (EER) and the control event rate (CER). However, the measurement of NNT needs to have a specific event. Therefore, calculating the NNT from continuous data would encounter some difficulties.
  2. Restoring facial esthetic and occlusal function in patients with cleft-related maxillary hypoplasia can benefit from more significant forward advancement after maxillary distraction. In the present retrospective study, we evaluated the mean difference between linear and angular parameters and compared bone-borne RED with traditional tooth-borne RED regarding distraction effectiveness, blood loss, operative time, and long-term stability. Both the tooth-borne and bone-borne groups presented significant maxillary protraction results from the original retrognathic maxillary position. Therefore, it is difficult to set a cut-off point from our continuous data in NNT measurement. Besides, in our study, assuming normality for outcome assessment for calculating NNT measurement is challenging. Moreover, intentionally-untreated severe maxillary retrusion in patients with cleft lip and palate could highly involve an ethical issue in medical experimentation since the further worsening facial and jawbones development can significantly deteriorate the treatment outcome in the future and inevitably impact the psychosocial self-esteem of patients. Therefore, finding an untreated patient with cleft-related maxillary hypoplasia as a control group is complicated in previous literature and our study.
  3. Intention-to-treat analysis is a method for analyzing results in a prospective randomized study where all participants are randomized allocating to the treatment group or control group. However, the current study indicated that ITT analysis is also challenging to perform in the RCTs in orthodontic research because of the potential lack of understanding and quite tricky to manage dropouts and missing data during long treatment duration and follow-up period (Please refer to the Reference: Barretto Dos Santos Lopes Batista K; Thiruvenkatachari, B; O'Brien, K. Intention-to-treat analysis: Are we managing dropouts and missing data properly in research on orthodontic treatment? A systematic review. Am J Orthod Dentofacial Orthop. 2019, 155, 19–27). Furthermore, in this present retrospective study, we assessed the treatment outcomes of maxillary distraction by retrospective evaluation of the mean difference of measurements in both the tooth-borne RED group and bone-borne RED group.
  4. In the present study, the patients who met the inclusion criteria of this study were enrolled and allocated into two groups based on the surgical method employed. Not all the patients required bone-borne RED treatment. We still carry out the traditional tooth-borne RED method for those patients (1) who presented with inadequate bone quantity and quality to apply miniplates for maxillary distraction, (2) those patients’ willingness or desire to have the traditional tooth-borne RED method either for the economic issues or other factors. Therefore, it was improbable to allocate our patients without any conditioning randomly. Nevertheless, the bone-borne RED patients in this study were all growing patients with cleft-related maxillary hypoplasia, who were managed with our initial bone-borne RED protocol that was just developed. We had also tried to select the tooth-borne RED group of patients presenting similar skeletal deformities and requiring the same surgical modality so that the baseline of assessment could be compared. We have already attached the cephalometric analysis of initial skeletal discrepancies and surgical outcomes; hopefully these can give the readers more precise insights.
  5. Line – should be 3.3. S

Response:  We appreciate the reviewer for the helpful comment, and we have revised the Line 268 from ”3.3. s” to “3.3. S” in the Result part according to the comment of reviewer.

Below please find the changed parts of the revised manuscript:

Results
3.3. Surgical outcomes

Page 8
Line 270

  1. Were the evaluators blinded?

Response: We thank the reviewer for this comment. Please find below point-by-point responses to the reviewer’s comments.

  1. All the patients required tooth-borne RED and bone-borne RED approach for correcting cleft-related maxillary hypoplasia in this study were performed by the same craniofacial surgeon (J.P Lai) who has been an experienced consultant in craniofacial surgery. He currently is the Dean of the Craniofacial Center of Kaohsiung Chang Gang Memorial Hospital. He is also in charge of training programs in this field.
  2. A single examiner measured and recorded all linear and angular parameters assessments from each landmark on lateral cephalograms (C.Y Tsai). Figure 3 demonstrated the lateral cephalograms of bone-borne RED with a transcutaneous wire system and traditional tooth-borne RED with an intra-oral splint. The distraction appliance could be detected on lateral cephalograms; therefore, it was complicated and not feasible to blind the single evaluator from identifying the distraction method employed. However, we had carried out the error study to minimize the potential discrepancy in systemic and random error (Houston, 1993) during the process, such as clarifying landmark identification, error threshold in duplicated identification of landmarks, and computed calculation of linear and angular parameters derived from the selected landmarks. Therefore, the measurements in our study were considered reliable.
  1. References should be in journal style and updated, there are several references too old: ex  1992, 1997, 1999, 2001,1993

Response:  We appreciate the reviewer for the helpful comment. McCarthy JG, Figueroa AA, and Polley JW were influential researchers applying distraction osteogenesis to treat craniofacial deformities. They first introduced the techniques of mandibular and maxillary distraction, which provided classic and valuable works of literature and clinical insights. According to the reviewer’s comment, we have removed old references except for the studies conducted by the author mentioned above. We also revised the quotation of the journal in the Reference parts, which follow the reviewer's comments. Please refer to the Reference sections, which are highlighted in yellow.

  1. Please define the following in the manuscript: 

ANS (x), mm

ANS(y), mm

PNS (x), mm

PNS (y), mm

A(x), mm

A(y), mm

A-Nv, mm

U1 (x), mm

U1 (y), mm

U6 (x), mm

U6 (y), mm

Prn (x), m

Prn (y), mm

A_’ _(_x_)_,_ _m_m_ _

A_’ _(_y_)_,_ _m_m_ _

Response: We appreciate the reviewer for the helpful comment. We have included the definition of linear parameters in Table 1. The Table Number in the manuscript were also revised accordingly. Please refer to pages 6 and 7 of the revised manuscript.

Below please find the changed parts of the revised manuscript:

  1. Materials and Methods

2.6. Outcome assessment

Page 6

Line 223-224

Table 1 illustrated the definition of linear parameters in relation to the X-, Y-axis.

Page 7

Line 237

Table 1. Definition of the distance from the landmarks to the X-, Y-axis.

Parameters

Abbreviation

Definition

Linear parameters

ANS to X-axis, mm

ANS (x), mm

The distance between the ANS to the X-axis. Points inferior to the X-axis are given a positive value.

ANS to Y-axis, mm

ANS (y), mm

The distance between the ANS to the Y-axis. Points anterior to the Y-axis are given a positive value.

PNS to X-axis, mm

PNS (x), mm

The distance between the PNS to the X-axis. Points inferior to the X-axis are given a positive value.

PNS to Y-axis, mm

PNS (y), mm

The distance between the PNS to the Y-axis. Points anterior to the Y-axis are given a positive value.

A point to X-axis, mm

A(x), mm

The distance between the A point to the X-axis. Points inferior to the X-axis are given a positive value.

A point to Y-axis, mm

A(y), mm

The distance between the A point to the Y-axis. Points anterior to the Y-axis are given a positive value.

Upper incisor to X-axis, mm  

U1 (x), mm 

The distance between the central incisor crown tip to the X-axis. Points inferior to the X-axis are given a positive value.

Upper incisor to Y-axis, mm  

U1 (y), mm 

The distance between the central incisor crown tip to the Y-axis. Points anterior to the Y-axis are given a positive value.

Upper molar to X-axis, mm

U6 (x), mm 

The distance between the distal marginal ridge of first molar to the X-axis. Points inferior to the X-axis are given a positive value.

Upper molar to Y-axis, mm

U6 (y), mm 

The distance between the distal marginal ridge of first molar to the Y-axis. Points anterior to the Y-axis are given a positive value.

Pronasale to X-axis, mm

Prn (x), mm

The distance between the nasal tip to the X-axis. Points inferior to the X-axis are given a positive value.

Pronasale to Y-axis, mm

Prn (y), mm

The distance between the nasal tip to the Y-axis. Points anterior to the Y-axis are given a positive value.

A’ soft tissue point to X-axis, mm

A’ (x), mm

T he distance between the soft tissue A point to the X-axis. Points inferior to the X-axis are given a positive value.

A’ soft tissue point to Y-axis, mm

A’ (y), mm

The distance between the soft tissue A point to the Y-axis. Points anterior to the Y-axis are given a positive value.

A point to Nperp

A-Nv

The distance between the A point to the Nperp reference line. Points anterior to the Nperp are given a positive value, while posterior to Nperp are assigned a negative value

Reference lines

Frankfort horizontal plane

FH plane

Plane constructed by Po and Or

Nasion perpendicular line

Nperp

Nv Vertical reference to FH plane and passing through Nasion.

Round 2

Reviewer 1 Report

The authors revised the article precisely . Thank you for this opportunity.

Reviewer 2 Report

Dear authors, 

Congratulations on your work!

Now the manuscript is significantly improved and can be easily perused by readers. 

Wish you good luck in the future in helping many class III cleft patients.